# A Short Functional Neuroimaging Assay Using Attachment Scenes to Recruit Neural Correlates of Social Cognition—A Replication Study

**DOI:** 10.3390/brainsci12070855

**Published:** 2022-06-29

**Authors:** Karin Labek, Lisa Dommes, Julia Eva Bosch, Matthias Schurz, Roberto Viviani, Anna Buchheim

**Affiliations:** 1Institute of Psychology, Faculty of Psychology and Sport Science, University of Innsbruck, 6020 Innsbruck, Austria; matthias.schurz@uibk.ac.at (M.S.); roberto.viviani@uibk.ac.at (R.V.); anna.buchheim@uibk.ac.at (A.B.); 2Department of Psychiatry and Psychotherapy III, Ulm University Medical Center, 89075 Ulm, Germany; lisa.dommes@uni-ulm.de (L.D.); julia.bosch@uni-ulm.de (J.E.B.)

**Keywords:** attachment theory, fMRI, social cognition, mentalization, adult attachment projective picture system, replication

## Abstract

Attachment theory provides a conceptual framework to understand the impact of early child–caregiver experiences, such as loss or separation, on adult functioning and psychopathology. In the current study, scenes from the Adult Attachment Projective Picture System (AAP), a validated, commonly used standardized diagnostic instrument to assess adult attachment representations, were used to develop a short fMRI assay eliciting the neural correlates of encoding of potentially hurtful and threatening social situations such as social losses, rejections or loneliness. Data from healthy participants (N = 19) showed activations in brain areas associated with social cognition and semantic knowledge during exposure to attachment-related scenes compared to control scenes. Extensive activation of the temporal poles was observed, suggesting the use of semantic knowledge for generating social concepts and scripts. This knowledge may underlie our ability to explain and predict social interactions, a specific aspect of theory of mind or mentalization. In this replication study, we verified the effectiveness of a modified fMRI assay to assess the external validity of a previously used imaging paradigm to investigate the processing of emotionally negatively valenced and painful social interactions. Our data confirm the recruitment of brain areas associated with social cognition with our very short neuroimaging assay.

## 1. Introduction

Research on human attachment has provided evidence on the importance of the child–caregiver relationship in the development of personality functioning and in psychopathology (for reviews, see [1]) in concomitance with childhood experiences of separation and loss. Based on these childhood experiences, the adult attachment system establishes relatively stable mental representations (adult attachment representations) through which people evaluate, interpret and predict and react adequately within social interactions. Deficient co-regulation capacities during interactions with significant others and poor social communication in infancy are understood as precursors that promote deficits in emotion regulation and social cognition across development. Due to its predominance and relevance for human functioning associated with intimate and significant relationships in general, neuroscientists have attempted to investigate the underlying neuronal processes associated with the human attachment system in recent years [1,2,3,4,5,6,7].

An evolving research area has focused on the identification of cortical substrates of adult attachment representations using functional imaging [2]. The resulting imaging assays may be useful in characterizing brain processes involved with social interactions of negative valence (e.g., loneliness, social loss), thus providing valuable information on the nature of disorders in which attachment is thought to be part of the pathogenetic mechanism [3,8,9].

An issue raised in previous imaging studies concerns the cortical substrates associated to exposure to the drawings of the Adult Attachment Projective Picture System (AAP) [10], a validated diagnostic instrument for the individual assessment of attachment patterns. The drawings of the AAP represent theoretically motivated attachment-related scenes portraying one or two individuals in different situations, such as loneliness or separation in the context of mother–infant interaction, illness or loss. When contrasted to exposure to carefully matched neutral control pictures, the pictures from the AAP have been shown in a previous functional imaging study [7] to activate a set of cortical areas associated with social cognition, semantic memory of social knowledge and reasoning about mental states of others [11].

An issue emerging from these neuroimaging studies is the relationship with studies explicitly investigating the neural correlates of loss or mourning [12,13], which showed activation of cortical midline structures. A recent study employed scenes of mourning individuals similar stylistically and in composition to those of the AAP scenes [14], thus providing the opportunity to draw comparisons. This study elicited activations in somatosensory areas and in the adjacent parietal operculum, the posterior cingulus and precuneus and the superior temporal gyrus, all areas associated with representations of affect and social cognition [15]. The picture set of the mourning individuals in that study differed from the AAP scenes in that they represented specific mourning postures, thereby being more explicit in their subject matter [14]. However, relative to previous studies with the AAP scenes [7], it used a much more rapid presentation of the visual material [14].

The present study had three aims. First, as noted, we were interested in verifying the replicability of our previous study (AAP here). Second, we were interested in identifying the neural substrates that may be shared with those activated in studies of negative emotion and the substrates specific to the attachment-related scenes, as well as with studies of social cognition more generally. To this end, we applied a quantitative decoding approach by comparing our fMRI results with the data of the Neurosynth database (www.neurosynth.org; [16]) using the NeuroVault tool [17,18]. Third, we used the same short presentation paradigm of the mourning study [14]. Besides ensuring comparability, we aimed at verifying the effectiveness of the functional imaging probe at this extremely short duration (about 1.5 min). The potential benefit of a short paradigm is the simultaneous collection of data from several assays in the same fMRI session, each constituted by a different task. Especially in clinical batteries, short paradigms might be required to avoid the effects of fatigue.

## 2. Materials and Methods

### 2.1. Participants

The initial sample consisted of 21 healthy participants recruited through local announcements. Two participants interrupted the experiment while in the scanner. The final sample comprised n = 19 (14 females), and participants’ mean age was 24.16 years (std. dev. = 4.66). Depression severity was assessed using the computerized German Version of the Centre for Epidemiologic Studies Depression Scale (CES-D; German Version: CES-D [19]; German Version [20]). The present and long-lasting anxiety levels were measured by the State-Trait Anxiety Inventory (STAI [21]; German Version [22]). All participants had scores in a subclinical range. Finally, participants were assessed with the International Neuropsychiatric Interview (M.I.N.I. 5.0.0 [23]; German version [24]) to exclude previous psychiatric or psychological illnesses (see Table 1).

### 2.2. Stimulus Material

The stimulus set was taken from the Adult Attachment Projective Picture System (AAP), a semi-standard interview to assess adult attachment representations [10]. The original AAP consists of eight drawings: one “warm-up” scene and seven attachment-related scenes depicting individuals, alone or in pairs, depicting situations that may activate the attachment system (such as separation, death, loneliness, loss) across the life span.

The control condition included eight drawings of non-attachment-related scenes carefully matched with the attachment-related AAP drawings of the original set of pictures. The first seven control scenes were obtained by manually redrawing aspects of the original AAP scenes to replace the elements that would have activated the attachment system with neutral elements while preserving the spatial arrangement and the shape of the objects as closely as possible. For example, reaching out hands to be embraced in the AAP scene was replaced by outstretched hands manipulating an object (see Figure 1, bottom row A and B). The remaining control scene consisted of the warm-up scene of the AAP set. To obtain a corresponding attachment-related picture, we created an attachment scene equivalent to the AAP stimuli for the warm-up scene using the same detail redrawing procedure. A third scrambled picture set was designed as a baseline visual condition, in which randomly arranged lines replaced persons and objects. The scrambled pictures were aligned in the spatial structure so as to match the two other picture sets and had the same size. The final set consisted of three different picture sets: eight attachment-related scenes (seven from the original AAP set, and one drawn to match the AAP warm-up scene), eight neutral versions of the same scenes (seven drawn to match the original AAP scenes, and the original AAP warm-up scene) and sixteen scrambled pictures for the baseline condition. For more detailed information about the development and validation of the stimulus material, see Labek et al. [7].

Task presentation was implemented using the commercial software package Presentation (Presentation 14, Neurobehavioral Systems Inc., Albany, CA, USA). All data were acquired in a single run consisting of 4 blocks of scenes (subdivided into 2 blocks attachment-related, 2 blocks control). In each block, four scenes were shown consecutively for 3 s for a total duration of 12 s for each block. These blocks were separated by 4 blocks presenting scrambled scenes of the same duration. The whole experiment required 1 min 39 s, including a trail-off interval of 3 s, resulting in 50 scans per experiment.

### 2.3. MRI Data Acquisition, Preprocessing and Statistical Modeling

MR images were acquired using a 3 T Prisma system (Siemens, Erlangen, Germany) located on the premises of the Department of Psychiatry and Psychotherapy at the University of Ulm. Functional data were acquired using T2*-weighted echo-planar imaging sequence (EPI) sensitive to the blood oxygen level-dependent (BOLD) signal (TR/TE 1970/36 ms, slice thickness 2.5 mm with a gap of 0.75 mm, giving a voxel size of 3.00 × 3.00 × 3.25 mm, flip angle 90°, bandwidth 1776 Hz/pixel, 32 transversal slices, anterior-to-posterior phase encoding, FOV 192/174 mm in the frequency/phase encoding directions, giving an image size of 64 × 58 × 32 voxels).

Functional images were processed with SPM8 (Wellcome Centre for Human Neuroimaging, University College, London, UK, (www.fil.ion.ucl.ac.uk/spm, accessed on 1 March 2012) in MATLAB (Mathworks Inc., Sherborn, MA, USA). For each participant, data were realigned to the first image of the series and re-sampled to a voxel size of 2 × 2 × 2 mm and normalized to standard MNI space. They were subsequently spatially smoothed with an 8 mm full width at half-maximum (FWHM) Gaussian kernel.

Data analysis proceeded at the first level using the general linear model. The design matrix included two predictors representing the two conditions (attachment-related and neutral) and six regressors obtained from realignment to model head motion confounds. Estimates of the contrast image attachment-related vs. neutral from all participants were entered into a second-level analysis, where full brain activations were thresholded voxel-wise at *p* < 0.001 to define clusters for whole-brain family-wise error rate correction based on Gaussian random field theory. All significance levels reported in the manuscript were corrected for the whole volume at cluster or peak level.

From the technical point of view of the study design, the effectiveness of a short paradigm is made plausible by the fact that in a repeated measurements setting, variance ensues at two different levels: between scans and between subjects. Here, notwithstanding the brevity of the paradigm, there were fifty scans at the first level, which are multiplied by the number of subjects at the second level for estimates of the precision of between scans effects [25], suggesting sufficient precision of estimate parameters at this level. Overlays were produced with the freely available software MRIcron (http://people.cas.sc.edu/rorden/mricron/index.html, accessed on 11 September 2019).

### 2.4. Overlap with Other Functional Activation Maps

After the fMRI data analysis, we used the Neurosynth decoding tool to extract relevant topics generated from the resulting *t*-map of our main contrast from 19 participants. Neurosynth is a valuable open-source web-based database for human neuroimaging research that aggregates activation coordinates from tens of thousands of published fMRI studies and published papers and compiles thematic maps based on their spatial distribution [16]. The Neurosynth decoder function (https://neurosynth.org) uses text mining and meta-analysis techniques to create accurate mappings between the brain activation patterns and peak signal coordinates in the database with associated topics.

Taking a reverse inference approach, we first uploaded our main contrast *t*-statistics map (after removing negative coefficients) to NeuroVault [17,18]. NeuroVault is a repository for neuroimaging studies with an implemented interface to the web-based Neurosynth platform. After loading our activation map into NeuroVault, the Neurosynth decoder function computed a voxel-wise Pearson correlation coefficient between our statistical *t*-map and the term-based statistic maps extracted from Neurosynth database. This correlation was used by Neurosynth to create a ranked list of terms arising from previous studies activating the same regions. In the next step, all anatomical and methodological terms were excluded from this list (which always give high scores in the decoder, but are not informative about function). Two raters evaluated the resulting list and selected by consensus the most relevant hits from the Neurosynth terms, revealing the following domains: social cognition, the default mode network, memory, semantic and language/music.

## 3. Results

### 3.1. Attachment-Related Pictures vs. Neutral Pictures

The contrast attachment vs. neutral revealed significant activations in associative areas such as the right superior temporal gyrus (Table 2, cluster #1) extending to parts of the middle temporal gyrus, the supramarginal gyrus and Rolandic operculum. A similar effect was observed on the left side, with activation peaks located in the supramarginal gyrus involving the superior and middle temporal (cluster #3) and angular gyri (cluster #4). These temporoparietal clusters also extended into parts of the Rolandic operculum (involving Heschl’s gyrus on its course), and spread anteriorly to the temporal poles. Additionally, there was a significant bilateral activation in the fusiform gyrus (cluster #2) extending at trend level into the parahippocampal gyrus (Figure 2).

The right medial portion of the same contrast showed a significantly greater activation in the middle/posterior cingulus (cluster #5) and in the adjacent cuneus/precuneus (cluster #6).

In the contrast control vs. attachment, there was a significant effect on the right hemisphere in the primary visual cortex (calcarine cortex, cluster #7).

### 3.2. Decoding Analysis

To provide valid reverse inference into the mental and psychological processes, we performed functional decoding of the results of the main contrast of our fMRI data from the entire sample. We used the Neurosynth decoder function to assess the similarity of the activation of the non-thresholded result map of our main contrast without negative values generated for the entire set of terms included in that database [16].

Decoding analysis revealed that the regions identified by the map of the main contrast, attachment vs. control, had two main areas of association (Figure 3). One referred to auditory processing and language (language, music, speech, voice). A second cluster included terms referring to social cognition (social, mental states, mentalizing, theory mind). Other terms retrieved by this analysis, such as semantic memory and default mode network, may refer to auditory and supramodal association areas [15,26], respectively, which were active in this contrast.

## 4. Discussion

Exposure to attachment-related images was associated in our study with a cluster of bilateral activations in temporoparietal areas (supramarginal and angular gyrus, the superior and middle temporal gyrus and, more ventrally, the fusiform gyrus). More anteriorly, the temporal poles were also active. A second group of activations was detected in the medial portion of the brain (posterior cingulate and precuneus). These areas are predominantly described in the literature as being of associative nature, and include neural substrates of social cognition as reported in previous studies of encoding of human actions, feelings or postures from exposure to attachment-related scenes [3,11,14,27,28]. An important issue that is debated in the literature is the differentiation of these substrates into correlates of specific aspects of social cognition or different forms of empathy [11,15,29,30]. This is an issue of primary importance for clinical and psychotherapy research due to the relevance of deficits in specific forms of empathy or social cognition in patients with psychiatric disorders, as documented in functional imaging studies of borderline personality disorders and depression [3,31,32] and their changes after psychotherapy [9].

When considering previous studies concerned with representations of attachment more specifically, we found correspondence between the areas detected here and those reported in the literature. A recently published fMRI study by [33] examined brain responses to pictorial representations of attachment separation in children (9–11 years), reporting increased activity in the precuneus/posterior cingulate, the superior/middle temporal gyri and the temporo-parietal junction. Similar activations in the medial prefrontal cortex, the posterior cingulate and precuneus, the middle temporal gyrus and anterior temporal poles have already been found in previous studies using the same scenes from the AAP compared to neutral control images [7] and in a single case study together with personalized attachment sentences vs. control sentences [34]. Some of the cortical areas detected in the comparison between attachment-related images in the present study (the middle and superior temporal gyri and the adjacent Rolandic operculum, and medial the posterior cingulate and precuneus) overlapped with the effect of exposure to mourning images [14]. However, each of these areas has also been involved in studies of social cognition not specifically concerned with attachment issues.

The parietal operculum has been shown in FMRI studies and in studies using electrical stimulation to host the secondary somatosensory cortex, SII [35,36,37,38]. Accordingly, recent studies looking at cytoarchitectonic parcellations have found reliable evidence that the parietal operculum in particular plays a key role in processing painful, somatosensory, auditory, interoceptive and stimuli relevant in encoding empathy tasks [15,39,40]. Another group of activations, located in the medial posterior area of the brain (precuneus and posterior cingulate), have been identified in studies on autobiographical/episodic memory, grief and pain [12,13,14,41,42,43,44].

The middle and superior temporal gyri have been shown to be involved in encoding visual features of bodies and bodily motion [14] and neuropsychological studies of patients with cortical lesions [45]. The activation of this cluster of related areas in the present study is therefore consistent with their reported role in encoding visual features of stimuli relevant to social cognition tasks [29,39,45,46,47]. In the present study, we also identified activity in the temporo-parietal junction/angular gyrus and in parts of the anterior temporal lobe (anterior temporal poles, fusiform gyrus). The right temporoparietal junction has been described as a core area in numerous studies with theory of mind tasks and mentalization. Activations in the angular gyrus are thought to support access to mental representations of others through perspective taking [11,48,49,50]. In addition, activations of the TPJ and anterior temporal poles have been associated with semantic categorization in studies investigating social cognition and expressive body posture relevant to social functioning [48,49].

Relative to a study of exposure to explicit mourning scenes [14], which used a similar paradigm, a characteristic activation of the present study concerned the involvement of the temporal poles. Findings in the neuroimaging literature provide converging evidence that activations in the anterior temporal poles are associated with the retrieval of abstract semantic concepts encoding social scripts or interactions [11,51,52]. It has been suggested that these scripts might play an essential role in the interpretation of situations or stories irrespective of the involvement of mentalization [11]. This role is confirmed in studies investigating deficits in social cognition in patients with cortical lesions in this area [53,54]. Brain lesion studies in the anterior temporal lobe also showed dysfunctions in retrieving representations of relationships that navigate social interactions [55,56] and representations of the self [57,58,59]. The involvement of the temporal poles in the present study may therefore be associated with recruitment of processes involved in the analysis of social interactions occurring in the AAP scenes. In addition to function within a particular psychological domain such as social cognition, there are also findings of a more general contribution the anterior temporal lobe makes to semantic processing [45,60].

Schurz and colleagues (2021) recently published a meta-analysis of neuroimaging studies of social cognition and empathy, in which they provide evidence for a model of social cognition in a multilevel approach with a hierarchical structure. Their data suggest the existence of three groups of processes involved in social cognition: cognitive, affective and a combination of both cognitive and affective processes. The cognitive cluster from their meta-analysis [15] revealed activations in the midline cortical cortex (precuneus and parts of the midcingulate) and temporoparietal areas, medial prefrontal areas and additionally in the anterior temporal cortices, which correspond, except for the activations in the parietal operculum (somatosensory), to the results of the present study. In contrast, there was little overlap between our activation patterns and the emotional cluster identified in their meta-analysis. Furthermore, meta-analytic decoding of the identified neural patterns identified social cognition terms, such as mentalizing, mental states and Theory of Mind [15]. An unexpected finding of this analysis was the retrieval of terms related to auditory encoding attributable to the recruitment of the posterior portion of the superior temporal gyrus. The co-activation of theory of mind tasks and verbal/auditory tasks near Heschl’s gyrus has been noted in systematic reviews of auditory areas [61].

These findings suggest that attachment-related material prevalently recruited cortical areas associated with cognitive components of social cognition. Similar activations (precuneus, temporoparietal junction area and medial superior frontal gyrus) described as a mentalization network were also prevalent in the findings of a study in a sample of children exposed to the AAP scenes, where areas associated with emotional processing were detected only as individual differences among attachment classifications [33]. In recent theoretical and empirical conceptualizations of attachment, both the attachment system and the mentalizing or social cognition system are closely interconnected and provide central abilities essential to social functioning [6,62]. Given that human attachment is a complex phenomenon, our findings underscore the interplay between social cognitive substrates and those involved with painful affective processing related to loss that may be relevant in its function in healthy adults when confronted with attachment-related material.

The present study also addresses the issue of reproducibility and replicability of our paradigm and provides information on the effectiveness of a short fMRI assay. The importance of verifying replicability of neuroimaging effects is being increasingly recognized, especially in the context of prospective clinical applications [63]. The potential advantages of a short paradigm accrue at different levels: less time spent in the scanner (often a burdensome time for participants), avoidance of habituation and fatigue in the task (thus increasing its validity) and the possibility of the simultaneous and more comprehensive assessment of diverse brain–behavior circuits in combination with other fMRI assays.

Several limitations of the present study should also be considered. Notwithstanding replication of group effects in these studies, it is important to stress that in clinical applications, replication of individual differences would be important. We could not assess this aspect of replication here since the participants differed between the present and previous studies. Furthermore, estimates of sample sizes to achieve reliable results in such individual effects would also be required.

In summary, exposure to the AAP scenes with this study design elicited the activation of neural substrates of the cognitive component of social cognition, including those concerned with the encoding of mental states of others, and of the Rolandic/parietal operculum and precuneus, which may relate to encoding painful experiences such as those related to loss. This novel paradigm may be of use in composite fMRI batteries for the assessment of intermediate phenotypes referring to the neural processing of social interactions with negative emotional valence.

## Figures and Tables

**Figure 1 brainsci-12-00855-f001:**
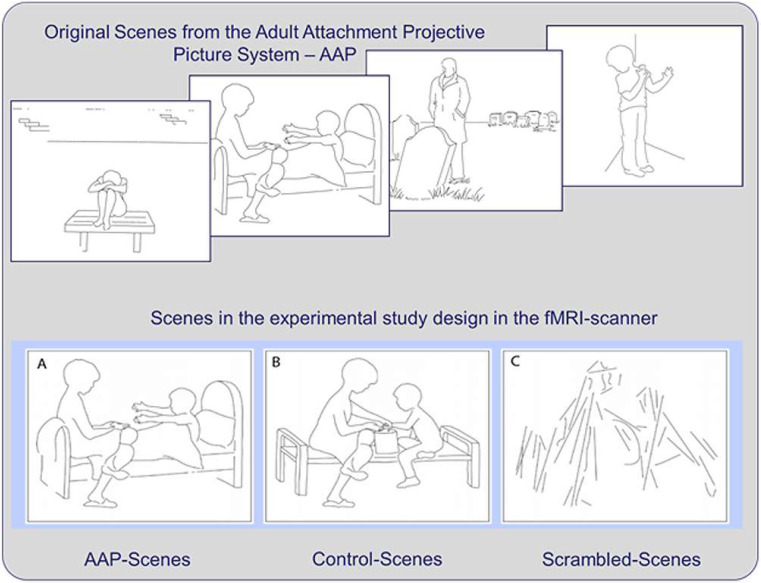
Illustration of four examples of the attachment scenes (top) (© George and West, 2012; all rights reserved). Example of a scene in the attachment version (**A**), the control version (**B**) and the scrambled scene version (**C**) (bottom).

**Figure 2 brainsci-12-00855-f002:**
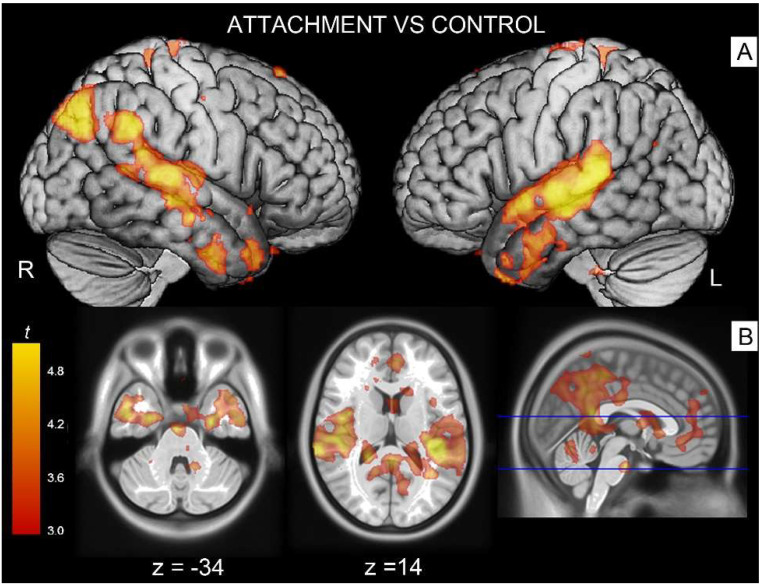
The red/orange areas illustrate activations elicited by the contrast attachment scenes vs. control stimuli (**A**) in the angular gyrus/superior temporal gyrus on the left side and the superior temporal gyrus on the right (**B**) spreading into the temporal poles. Activations shown as parametric maps of *t*-values overlaid on a template T1-weigthed brain. For illustration purposes, statistical maps were thresholded at *p* = 0.005 uncorrected.

**Figure 3 brainsci-12-00855-f003:**
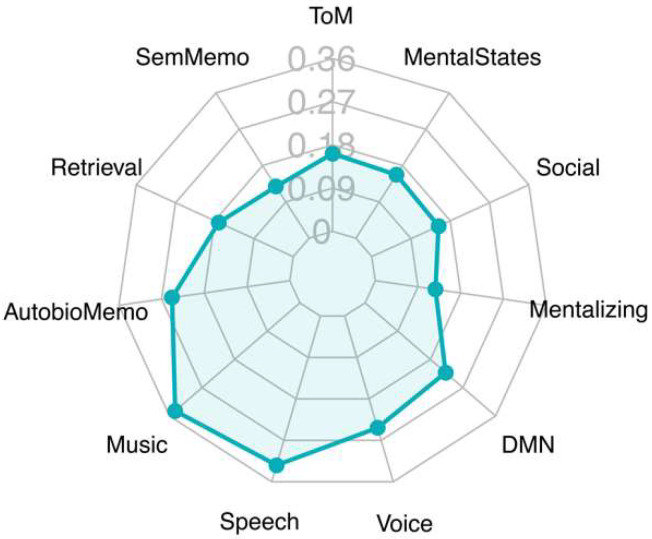
Radar chart showing most relevant topics of the Pearson correlations generated between statistical *t*-map of our main results and the Neurosynth term-based reverse inference activation maps database (www.neurosynth.org [16]). The decoder was used to compare the non-threshold statistical *t*-maps (negative values not included) with the statistical maps automatically generated by the Neurosynth database. Depicted above are the extracted topics together with the coefficient values. The results with the highest convergence (Pearson’s r) are shown. Abbreviations: AutobioMemo, autobiographical memory; DMN, default mode network; SemMemo, semantic memory; ToM, Theory of Mind.

**Table 1 brainsci-12-00855-t001:** The following table shows the description of the sample (age, gender) and results of the clinical measurements.

	n = 19
Age mean (std. dev.)	24.16 (4.66)
female (%)	14 (66.7)
male (%)	5 (23.8)
ADS mean (std. dev.)	8.84 (6.59)
STAI-S mean (std. dev.)	37.89 (8.77)
STAI-T mean (std. dev.)	39.21 (7.18)

**Table 2 brainsci-12-00855-t002:** Activation elicited by the contrast attachment vs. control stimuli.

		Cluster Level		Peak Level		
Cluster #	Region (Side)	Voxel	*p*	*t*	*p*	MNI Coordinates
		Count	*Clust*		*Peak*	(*x*, *y*, *z*)
		(*k*)				
	*Contrast Attachment* > *Neutral*					
1	Superior Temporal Gyrus (R)	3534	<0.001	8.09	0.013	52, −26, 4
	Thalamus (R)			7.24	0.043	54, −18, 0
	Parahippocampal Gyrus			6.81	0.079	20, −10, −22
2	Fusiform Gyrus (L)	498	<0.001	7.5	0.029	−28, −32, −16
3	Superior Temporal Gyrus/(L)	2313	<0.001	7.28	0.04	−44, −32, 10
	Supramarginal Gyrus (L)			6.15	0.204	−58, −48, 32
	Mid Temp Gyrus			5.82	0.318	−62, −30, −4
4	Angular Gyrus (L)	301	0.009	5.2	0.643	−44, −74, 44
5	Post Cingulum/Precuneus	217	0.036	4.34	0.982	12, −40, 10
6	Mid Cingulum/Precuneus (R)	180	0.069	4.97	0.772	8, −38, 52
	Mid Cingulum (L)			4.7	0.896	−8, −30, 52
	*Contrast Neutral > Attachment*					
7	Lingual/Calcarine (R)	357	0.004	5.29	0.589	8, −82, −8
	Calcarine (R)			4.57	0.939	18, −78, −12

Note: MNI coord: Montral Neurological Institute coordinates (in mm); *k*: cluster extent (in voxel of isotropic size, 2 mm); *p Clust*: significance level, family-wise error rate (FWE) cluster-level correction for the whole volume; *t*: Student’s t; *p Peak*: significance level, family-wise error rate (FWE) peak-level correction; Post: posterior; Mid: middle.

## Data Availability

Data are available from the corresponding author on request pending confirmation on conformity with the declared aims of the study in the informed consent forms of participants.

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
