# Peer review of "A Short Functional Neuroimaging Assay Using Attachment Scenes to Recruit Neural Correlates of Social Cognition—A Replication Study"

_brainsci, 2022, doi:10.3390/brainsci12070855_

Round 1

Reviewer 1 Report

The authors performed a fMRI study in an attempt to replicate their previous findings about the neural correlates of social cognition and to answer some question that poped up about their previous findings and those of others using other types of stimuli.

The study seems to be well designed and executed and the paper is well written. However, I have some remarks to make:

  • Table 1: What do the authors mean by ‘female mean (%)’ and ‘male mean (%)’? It looks like the authors report the number of females and males, but that is not a mean and the percentages does not correspond with the total sample size (n=19).
  • Why was slice time correction for the fMRI data not done in this study?
  • At the end of paragraph 2.3, the authors state that the short paradigm is effective by the multiplication of the number of scans and the number of subjects. However, only 1 contrast map is determined per subject, which are then used in the second level analysis. As such, you can not simply multiply the number of scans by the number of subjects for the effectiveness or power of your study. there are 2 levels of power to consider. At first, you have the power to detect the activation contrast per subject which is defined by the number of stimuli per condition in combination with the sampling of the hemodynamic response. Secondly, you have the power of your groups analysis which is defined by the number of subjects in your study.  Given the number of stimuli per condition (8) used in the short paradigm and the small number of subjects (19) in the study, both powers seems to be low. Can the authors comment on this?
  • Paragraph 2.4 How the identification of the terms is done is not completely clear to me. Especially the sentence ‘The terms corresponding to the activation pattern from the meta-analysis was determined by identifying the topic terms given by the activation of our main contrast statistic.’ is confusing.
  •  

Author Response

We would like to thank reviewer 1 for the comments and the editor for the opportunity to reply to this assessment of our manuscript. Please see the attachment.

Reviewer 2 Report

This fMRI study tested 19 neurotypical adults to replicate the use of a short scanning protocol using scenes from the Adult Attachment Projective Picture System (AAP) to examine brain activation patterns associated with attachment and social cognition in general. The results showed consistent findings with previous reports. I have some suggestions for the authors to consider. 

1. Overall, the objectives are clearly articulated in the text, and the analysis procedures and results are presented properly. However, for a replication study, it is important to recognize that consistency in group-level overall activations may not be an indicator of consistency for individual regions of interest and that functional interpretation of the brain activation in selected ROIs may not be interpreted literally across studies without understanding the actual relationships between neural and behavioral measures of social and affective cognition at the individual subject level. 

2. Qualitative assessment of activation patterns across studies is not a substitute of statistically robust procedures to assess the reliability and effect size in relation to sample size. If possible, some form of estimation of sufficient sample size to achieve reliable results with desirable effect size using the fast scan protocol is needed for both group-level analysis and individual-level analysis.

3. As no clinical comparison group was tested, replication of fMRI group activations does not imply replicable application of the protocol to various clinical populations. 

4. As fMRI activations are task-driven, replicability of fMRI results using the AAP stimuli in different tasks remains unclear. This may need to be acknowledged. 

references:

https://doi.org/10.1016/j.neuroimage.2019.116223

https://doi.org/10.1038/s42003-018-0073-z

https://doi.org/10.1016/j.neuroimage.2019.116223

Author Response

We would like to thank reviewer 2 for the comments and the editor for the opportunity to reply to this assessment of our manuscript. Please see the attachment.
